# On Training Implicit Models

**Zhengyang Geng**[1,2]* **Xin-Yu Zhang**[2]* **Shaojie Bai**[4] **Yisen Wang**[2,3] **Zhouchen Lin**[2,3,5]†

[1]Zhejiang Lab, China    [2]Key Lab. of Machine Perception, School of AI, Peking University
[3]Institute for Artificial Intelligence, Peking University
[4]Carnegie Mellon University    [5]Pazhou Lab, China

## Abstract

This paper focuses on training implicit models of infinite layers. Specifically, previous works employ implicit differentiation and solve the exact gradient for the backward propagation. However, *is it necessary to compute such an exact but expensive gradient for training?* In this work, we propose a novel gradient estimate for implicit models, named *phantom gradient*, that 1) forgoes the costly computation of the exact gradient; and 2) provides an update direction empirically preferable to the implicit model training. We theoretically analyze the condition under which an ascent direction of the loss landscape could be found, and provide two specific instantiations of the phantom gradient based on the damped unrolling and Neumann series. Experiments on large-scale tasks demonstrate that these lightweight phantom gradients significantly accelerate the backward passes in training implicit models by roughly $1.7\times$, and even boost the performance over approaches based on the exact gradient on ImageNet.

## 1 Introduction

Conventional neural networks are typically constructed by explicitly stacking multiple linear and non-linear operators in a feed-forward manner. Recently, the implicitly-defined models [1, 2, 3, 4, 5] have attracted increasing attentions and are able to match the state-of-the-art results by explicit models on several vision [3, 6], language [2] and graph [4] tasks. These works treat the evolution of the intermediate hidden states as a certain form of dynamics, such as fixed-point equations [2, 3] or ordinary differential equations (ODEs) [1, 7], which represents infinite latent states. The forward passes of implicit models are therefore formulated as solving the underlying dynamics, by either black-box ODE solvers [1, 7] or root-finding algorithms [2, 3]. As for the backward passes, however, directly differentiating through the forward pass trajectories could induce a heavy memory overhead [8, 9]. To this end, researchers have developed memory-efficient backpropagation via implicit differentiation, such as solving a Jacobian-based linear fixed-point equation for the backward pass of deep equilibrium models (DEQs) [2], which eventually makes the backpropagation trajectories independent of the forward passes. This technique allows one to train implicit models with essentially constant memory consumption, as we only need to store the final output and the layer itself without saving any intermediate states. However, in order to estimate the exact gradient by implicit differentiation, implicit models have to rely on expensive black-box solvers for backward passes, *e.g.,* ODE solvers or root-solving algorithms. These black-box solver usually makes the gradient computation very costly in practice, even taking weeks to train state-of-the-art implicit models on ImageNet [10] with 8 GPUs.

This work investigates fast approximate gradients for training implicit models. We found that a first-order oracle that produces good gradient estimates is enough to efficiently and effectively train implicit

---

*Equal contribution
†Corresponding author, zlin@pku.edu.cn

35th Conference on Neural Information Processing Systems (NeurIPS 2021).

models, circumventing laboriously computing the exact gradient as in prior arts [2, 3, 4, 11, 12]. We develop a framework in which a balanced trade-off is made between the precision and conditioning of the gradient estimate. Specifically, we provide the general condition under which the phantom gradient can provide an ascent direction of the loss landscape. We further propose two instantiations of phantom gradients in the context of DEQ models, which are based on the the damped fixed-point unrolling and the Neumann series, respectively. Importantly, we show that our proposed instantiations satisfy the theoretical condition, and that the stochastic gradient descent (SGD) algorithm based on the phantom gradient enjoys a sound convergence property as long as the relevant hyperparameters, *e.g.,* the damping factor, are wisely selected. Note that our method only affects, and thus accelerates, the backward formulation of the implicit models, leaving the forward pass formulation (*i.e.,* the root-solving process) and the inference behavior unchanged so that our method is applicable to a wide range of implicit models, forward solvers, and inference strategies. We conduct an extensive set of synthetic, ablation, and large-scale experiments to both analyze the theoretical properties of the phantom gradient and validate its speedup and performances on various tasks, such as ImageNet [10] classification and Wikitext-103 [13] language modeling.

Overall, our results suggest that: 1) the phantom gradient estimates an ascent direction; 2) it is applicable to large-scale tasks and is capable of achieving a strong performance which is comparable with or even better than that of the exact gradient; and 3) it significantly shortens the total training time needed for implicit models roughly by a factor of $1.4 \sim 1.7\times$, and even accelerates the backward passes by astonishingly $12\times$ on ImageNet. We believe that our results provide strong evidence for effectively training implicit models with the lightweight phantom gradient.

## 2 Method

### 2.1 Inspection of Implicit Differentiation

In this work, we primarily focus on the formulation of implicit models based on root-solving, represented by the DEQ models [2]. The table of notations is arranged in Appendix A. Specifically, given an equilibrium module $\mathcal{F}$, the output of the implicit model is characterized by the solution $\boldsymbol{h}^*$ to the following fixed-point equation:

$$\boldsymbol{h}^* = \mathcal{F}(\boldsymbol{h}^*, \boldsymbol{z}), \tag{1}$$

where $\boldsymbol{z} \in \mathbb{R}^{d_{\boldsymbol{u}}+d_{\boldsymbol{\theta}}}$ is the union of the module's input $\boldsymbol{u} \in \mathbb{R}^{d_{\boldsymbol{u}}}$ and parameters $\boldsymbol{\theta} \in \mathbb{R}^{d_{\boldsymbol{\theta}}}$, *i.e.,* $\boldsymbol{z}^\top = [\boldsymbol{u}^\top, \boldsymbol{\theta}^\top]$. Here, $\boldsymbol{u}$ is usually a projection of the original data point $\boldsymbol{x} \in \mathbb{R}^{d_{\boldsymbol{x}}}$, *e.g.,* $\boldsymbol{u} = \mathcal{M}(\boldsymbol{x})$. In this section, we assume $\mathcal{F}$ is a contraction mapping *w.r.t.* $\boldsymbol{h}$ so that its Lipschitz constant $L_{\boldsymbol{h}}$ *w.r.t.* $\boldsymbol{h}$ is less than one, *i.e.,* $L_{\boldsymbol{h}} < 1$, a setting that has been analyzed in recent works [14, 15][1].

To differentiate through the fixed point by Eq. (1), we need to calculate the gradient of $\boldsymbol{h}^*$ *w.r.t.* the input $\boldsymbol{z}$. By Implicit Function Theorem (IFT), we have

$$\frac{\partial \boldsymbol{h}^*}{\partial \boldsymbol{z}} = \frac{\partial \mathcal{F}}{\partial \boldsymbol{z}}\bigg|_{\boldsymbol{h}^*} \left(\boldsymbol{I} - \frac{\partial \mathcal{F}}{\partial \boldsymbol{h}}\bigg|_{\boldsymbol{h}^*}\right)^{-1}. \tag{2}$$

Here, $(\partial \boldsymbol{a}/\partial \boldsymbol{b})_{ij} = \partial \boldsymbol{a}_j/\partial \boldsymbol{b}_i$. The equilibrium point $\boldsymbol{h}^*$ of Eq. (1) is then passed to a post-processing function $\mathcal{G}$ to obtain a prediction $\hat{\boldsymbol{y}} = \mathcal{G}(\boldsymbol{h}^*)$. In the generic learning scenario, the training objective is the following expected loss:

$$\mathcal{R}(\boldsymbol{\theta}) = \mathbb{E}_{(\boldsymbol{x},\boldsymbol{y}) \sim \mathcal{P}} \left[\mathcal{L}(\hat{\boldsymbol{y}}(\boldsymbol{x}; \boldsymbol{\theta}), \boldsymbol{y})\right], \tag{3}$$

where $\boldsymbol{y}$ is the groundtruth corresponding to the training example $\boldsymbol{x}$, and $\mathcal{P}$ is the data distribution. Here, we omit the parameters of $\mathcal{G}$, because given the output $\boldsymbol{h}^*$ of the implicit module $\mathcal{F}$, training the post-processing part $\mathcal{G}$ is the same as training explicit neural networks. The most crucial component is the gradient of the loss function $\mathcal{L}$ *w.r.t.* the input vector $\boldsymbol{z}^\top = [\boldsymbol{u}^\top, \boldsymbol{\theta}^\top]$, which is used to train both the implicit module $\mathcal{F}$ and the input projection module $\mathcal{M}$. Using Eq. (2) with the condition $\boldsymbol{h} = \boldsymbol{h}^*$, we have

$$\frac{\partial \mathcal{L}}{\partial \boldsymbol{u}} = \frac{\partial \mathcal{F}}{\partial \boldsymbol{u}} \left(\boldsymbol{I} - \frac{\partial \mathcal{F}}{\partial \boldsymbol{h}}\right)^{-1} \frac{\partial \mathcal{L}}{\partial \boldsymbol{h}}, \quad \frac{\partial \mathcal{L}}{\partial \boldsymbol{\theta}} = \frac{\partial \mathcal{F}}{\partial \boldsymbol{\theta}} \left(\boldsymbol{I} - \frac{\partial \mathcal{F}}{\partial \boldsymbol{h}}\right)^{-1} \frac{\partial \mathcal{L}}{\partial \boldsymbol{h}}. \tag{4}$$

The gradients in Eq. (4) are in the same form *w.r.t.* $\boldsymbol{u}$ and $\boldsymbol{\theta}$. Without loss of generality, we only discuss the gradient *w.r.t.* $\boldsymbol{\theta}$ in the following sections.

---

[1]Note that the contraction condition could be largely relaxed to the one that the spectral radius of $\partial \mathcal{F}/\partial \boldsymbol{h}^*$ on the given data is less than 1, *i.e.,* $\rho(\partial \mathcal{F}/\partial \boldsymbol{h}^*) < 1$, as indicated by the well-posedness condition in [5].

## 2.2 Motivation

The most intriguing part lies in the Jacobian-inverse term, *i.e.,* $(\boldsymbol{I} - \partial\mathcal{F}/\partial\boldsymbol{h})^{-1}$. Computing the inverse term by brute force is intractable due to the $\mathcal{O}(n^3)$ complexity. Previous implicit models [2] approach this by solving a linear system involving a Jacobian-vector product iteratively via a gradient solver, introducing over 30 Broyden [16] iterations in the backward pass. However, the scale of the Jacobian matrix can exceed $10^6 \times 10^6$ in the real scenarios, leading to a prohibitive cost in computing the exact gradient. For example, training a small-scale state-of-the-art implicit model on ImageNet can consume weeks using 8 GPUs while training explicit models usually takes days, demonstrating that pursuing the exact gradient severely slows down the training process of implicit models compared with explicit models.

Secondly, because of the inversion operation, we cast doubt on the conditioning of the gradient and the stability of training process from the numerical aspect. The Jacobian-inverse can be numerically unstable when encountering the ill-conditioning issue. The conditioning problem might further undermine the training stability, as studied in the recent work [17].

Plus, the inexact gradient [9, 18, 19, 20, 21, 22] is widely applied in the previous learning protocol, like linear propagation [23] and synthetic gradient [24]. Here, the Jacobian-inverse is used to calculate the exact gradient which is not always optimal for model training. Moreover, previous research has used a moderate gradient noise as a regularization approach [25], which has been shown to play a central role in escaping poor local minima and improving generalization ability [26, 27, 28].

The concerns and observations motivate us to rethink the possibility of replacing the Jacobian-inverse term in the standard implicit differentiation with a cheaper and more stable counterpart. We believe that an exact gradient estimate is not always required, especially for a black-box layer like those in the implicit models. Hence this work designs an inexact, theoretically sound, and practically efficient gradient for training implicit models under various settings. We name the proposed gradient estimate as the *phantom gradient*.

Suppose the Jacobian $\partial\boldsymbol{h}^*/\partial\boldsymbol{\theta}$ is replaced with a matrix $\boldsymbol{A}$, and the corresponding phantom gradient is defined as

$$\widehat{\frac{\partial\mathcal{L}}{\partial\boldsymbol{\theta}}} := \boldsymbol{A}\,\frac{\partial\mathcal{L}}{\partial\boldsymbol{h}}. \tag{5}$$

Next, we give the general condition on $\boldsymbol{A}$ so that the phantom gradient can be guaranteed valid for optimization (Sec. 2.3), and provide two concrete instantiations of $\boldsymbol{A}$ based on either damped fixed-point unrolling or the Neumann series (Sec. 2.4). The proofs of all our theoretical results are presented in Appendix C.

## 2.3 General Condition on the Phantom Gradient

Previous research on theoretical properties for the inexact gradient include several aspects, such as the gradient direction [22], the unbiasedness of the estimator [29], and the convergence theory of the stochastic algorithm [19, 30]. The following theorem formulates a sufficient condition that the phantom gradient gives an ascent direction of the loss landscape.

**Theorem 1.** *Suppose the exact gradient and the phantom gradient are given by Eq.* (4) *and* (5), *respectively. Let $\sigma_{max}$ and $\sigma_{min}$ be the maximal and minimal singular value of $\partial\mathcal{F}/\partial\boldsymbol{\theta}$. If*

$$\left\| \boldsymbol{A}\left(\boldsymbol{I} - \frac{\partial\mathcal{F}}{\partial\boldsymbol{h}}\right) - \frac{\partial\mathcal{F}}{\partial\boldsymbol{\theta}} \right\| < \frac{\sigma_{min}^2}{\sigma_{max}}, \tag{6}$$

*then the phantom gradient provides an ascent direction of the function $\mathcal{L}$, i.e.,*

$$\left\langle \widehat{\frac{\partial\mathcal{L}}{\partial\boldsymbol{\theta}}}, \frac{\partial\mathcal{L}}{\partial\boldsymbol{\theta}} \right\rangle > 0. \tag{7}$$

**Remark 1.** Suppose only the $(\boldsymbol{I} - \partial\mathcal{F}/\partial\boldsymbol{h})^{-1}$ term is replaced with a matrix $\boldsymbol{D}$, namely, $\boldsymbol{A} = (\partial\mathcal{F}/\partial\boldsymbol{\theta})\,\boldsymbol{D}$. Then, the condition in (6) can be reduced into

$$\left\| \boldsymbol{D}\left(\boldsymbol{I} - \frac{\partial\mathcal{F}}{\partial\boldsymbol{h}}\right) - \boldsymbol{I} \right\| < \frac{1}{\kappa^2}, \tag{8}$$

where $\kappa$ is the condition number of $\partial\mathcal{F}/\partial\boldsymbol{\theta}$. (See Appendix C.1 for the derivation.)

## 2.4 Instantiations of the Phantom Gradient

In this section, we present two practical instantiations of the phantom gradient. We also verify that the general condition in Theorem 1 can be satisfied if the hyperparameters in our instantiations are wisely selected.

Suppose we hope to differentiate through implicit dynamics, *e.g.*, either a root-solving process or an optimization problem. In the context of hyperparameter optimization (HO), previous solutions include differentiating through the unrolled steps of the dynamics [19] or employing the Neumann series of the Jacobian-inverse term [9]. In our case, if we solve the root of Eq. (1) via the fixed-point iteration:

$$\boldsymbol{h}_{t+1} = \mathcal{F}(\boldsymbol{h}_t, \boldsymbol{z}), \quad t = 0, 1, \cdots, T-1, \tag{9}$$

then by differentiating through the unrolled steps of Eq. (9), we have

$$\frac{\partial \boldsymbol{h}_T}{\partial \boldsymbol{\theta}} = \sum_{t=0}^{T-1} \frac{\partial \mathcal{F}}{\partial \boldsymbol{\theta}}\bigg|_{\boldsymbol{h}_t} \prod_{s=t+1}^{T-1} \frac{\partial \mathcal{F}}{\partial \boldsymbol{h}}\bigg|_{\boldsymbol{h}_s}. \tag{10}$$

Besides, the Neumann series of the Jacobian-inverse $(\boldsymbol{I} - \partial \mathcal{F}/\partial \boldsymbol{h})^{-1}$ is

$$\boldsymbol{I} + \frac{\partial \mathcal{F}}{\partial \boldsymbol{h}} + \left(\frac{\partial \mathcal{F}}{\partial \boldsymbol{h}}\right)^2 + \left(\frac{\partial \mathcal{F}}{\partial \boldsymbol{h}}\right)^3 + \cdots. \tag{11}$$

Notably, computing the Jacobian $\partial \boldsymbol{h}^*/\partial \boldsymbol{\theta}$ using the Neumann series in (11) is equivalent to differentiating through the unrolled steps of Eq. (9) at the exact equilibrium point $\boldsymbol{h}^*$ and taking the limit of infinite steps [9].

*Without altering the root* of Eq. (1), we consider a damped variant of the fixed-point iteration:

$$\boldsymbol{h}_{t+1} = \mathcal{F}_\lambda(\boldsymbol{h}_t, \boldsymbol{z}) = \lambda \mathcal{F}(\boldsymbol{h}_t, \boldsymbol{z}) + (1-\lambda)\boldsymbol{h}_t, \quad t = 0, 1, \cdots, T-1. \tag{12}$$

Differentiating through the unrolled steps of Eq. (12), Eq. (10) is adapted as

$$\frac{\partial \boldsymbol{h}_T}{\partial \boldsymbol{\theta}} = \lambda \sum_{t=0}^{T-1} \frac{\partial \mathcal{F}}{\partial \boldsymbol{\theta}}\bigg|_{\boldsymbol{h}_t} \prod_{s=t+1}^{T-1} \left(\lambda \frac{\partial \mathcal{F}}{\partial \boldsymbol{h}}\bigg|_{\boldsymbol{h}_s} + (1-\lambda)\boldsymbol{I}\right). \tag{13}$$

The Neumann series of $(\boldsymbol{I} - \partial \mathcal{F}/\partial \boldsymbol{h})^{-1}$ is correspondingly adapted as

$$\lambda \left(\boldsymbol{I} + \boldsymbol{B} + \boldsymbol{B}^2 + \boldsymbol{B}^3 + \cdots\right), \quad \text{where} \quad \boldsymbol{B} = \lambda \frac{\partial \mathcal{F}}{\partial \boldsymbol{h}} + (1-\lambda)\boldsymbol{I}. \tag{14}$$

The next theorem shows that under mild conditions, the Jacobian from the damped unrolling in Eq. (13) converges to the exact Jacobian and the Neumann series in (14) converges to the Jacobian-inverse $(\boldsymbol{I} - \partial \mathcal{F}/\partial \boldsymbol{h})^{-1}$ as well.

**Theorem 2.** *Suppose the Jacobian $\partial \mathcal{F}/\partial \boldsymbol{h}$ is a contraction mapping. Then,*

*(i) the Neumann series in (14) converges to the Jacobian-inverse $(\boldsymbol{I} - \partial \mathcal{F}/\partial \boldsymbol{h})^{-1}$; and*

*(ii) if the function $\mathcal{F}$ is continuously differentiable w.r.t. both $\boldsymbol{h}$ and $\boldsymbol{\theta}$, the sequence in Eq. (13) converges to the exact Jacobian $\partial \boldsymbol{h}^*/\partial \boldsymbol{\theta}$ as $T \to \infty$, i.e.,*

$$\lim_{T \to \infty} \frac{\partial \boldsymbol{h}_T}{\partial \boldsymbol{\theta}} = \frac{\partial \mathcal{F}}{\partial \boldsymbol{\theta}}\bigg|_{\boldsymbol{h}^*} \left(\boldsymbol{I} - \frac{\partial \mathcal{F}}{\partial \boldsymbol{h}}\bigg|_{\boldsymbol{h}^*}\right)^{-1}. \tag{15}$$

However, as discussed in Sec. 2.2, it is unnecessary to compute the exact gradient with infinite terms. In the following context, we introduce two instantiations of the phantom gradient based on the finite-term truncation of Eq. (13) or (14).

**Unrolling-based Phantom Gradient (UPG).** In the unrolling form, the matrix $\boldsymbol{A}$ is defined as

$$\boldsymbol{A}_{k,\lambda}^{\text{unr}} = \lambda \sum_{t=0}^{k-1} \frac{\partial \mathcal{F}}{\partial \boldsymbol{\theta}}\bigg|_{\boldsymbol{h}_t} \prod_{s=t+1}^{k-1} \left(\lambda \frac{\partial \mathcal{F}}{\partial \boldsymbol{h}}\bigg|_{\boldsymbol{h}_s} + (1-\lambda)\boldsymbol{I}\right). \tag{16}$$

**Neumann-series-based Phantom Gradient (NPG).** In the Neumann form, the matrix $\boldsymbol{A}$ is defined as

$$\boldsymbol{A}_{k,\lambda}^{\text{neu}} = \lambda \left. \frac{\partial \mathcal{F}}{\partial \boldsymbol{\theta}} \right|_{\boldsymbol{h}^*} \left( \boldsymbol{I} + \boldsymbol{B} + \boldsymbol{B}^2 + \cdots + \boldsymbol{B}^{k-1} \right), \quad \text{where } \boldsymbol{B} = \lambda \left. \frac{\partial \mathcal{F}}{\partial \boldsymbol{h}} \right|_{\boldsymbol{h}^*} + (1 - \lambda) \boldsymbol{I}. \quad (17)$$

Note that both the initial point of the fixed-point iteration (*i.e.*, $\boldsymbol{h}_0$ in Eq. (16)) and the point at which the Neumann series is evaluated (*i.e.*, $\boldsymbol{h}^*$ in (17)) are the solution of the root-finding solver. (See Appendix B for implementation of the phantom gradient.)

According to Theorem 2, the matrix $\boldsymbol{A}$ defined by either Eq. (16) or (17) converges to the exact Jacobian $\partial \boldsymbol{h}^* / \partial \boldsymbol{\theta}$ as $k \to \infty$ for any $\lambda \in (0, 1]$. Therefore, by Theorem 2, the condition in (6) can be satisfied if a sufficiently large step $k$ is selected, since

$$\left\| \boldsymbol{A} \left( \boldsymbol{I} - \frac{\partial \mathcal{F}}{\partial \boldsymbol{h}} \right) - \frac{\partial \mathcal{F}}{\partial \boldsymbol{\theta}} \right\| \leq (1 + L_{\boldsymbol{h}}) \left\| \boldsymbol{A} - \frac{\partial \mathcal{F}}{\partial \boldsymbol{\theta}} \left( \boldsymbol{I} - \frac{\partial \mathcal{F}}{\partial \boldsymbol{h}} \right)^{-1} \right\|. \quad (18)$$

Next, we characterize the impact of the two hyperparameters, *i.e.*, $k$ and $\lambda$, on the precision and conditioning of $\boldsymbol{A}$. Take the NPG (Eq. (17)) as an example.

(i) On the precision of the phantom gradient,
- a large $k$ makes the gradient estimate more accurate, as higher-order terms of the Neumann series are included, while
- a small $\lambda$ slows down the convergence of the Neumann series because the norm $\|\boldsymbol{B}\|$ increases as $\lambda$ decreases.

(ii) On the conditioning of the phantom gradient,
- a large $k$ impairs the conditioning of $\boldsymbol{A}$ since the condition number of $\boldsymbol{B}^k$ grows exponentially as $k$ increases, while
- a small $\lambda$ helps maintain a small condition number of $\boldsymbol{A}$ because the singular values of $\partial \mathcal{F} / \partial \boldsymbol{h}$ are "smoothed" by the identity matrix.

In a word, a large $k$ is preferable for a more accurate $\boldsymbol{A}$, while a small $\lambda$ contributes to the well-conditioning of $\boldsymbol{A}$. Practically, these hyperparameters should be selected in consideration of a balanced trade-off between the precision and conditioning of $\boldsymbol{A}$. See Sec. 3 for experimental results.

## 2.5 Convergence Theory

In this section, we provide the convergence guarantee of the SGD algorithm using the phantom gradient. We prove that under mild conditions, if the approximation error of the phantom gradient is sufficiently small, the SGD algorithm converges to an $\epsilon$-approximate stationary point in the expectation sense. We will discuss the feasibility of our assumptions in Appendix C.3.

**Theorem 3.** *Suppose the loss function $\mathcal{R}$ in Eq. (3) is $\ell$-smooth, lower-bounded, and has bounded gradient almost surely in the training process. Besides, assume the gradient in Eq. (4) is an unbiased estimator of $\nabla \mathcal{R}(\boldsymbol{\theta})$ with a bounded covariance. If the phantom gradient in Eq. (5) is an $\epsilon$-approximation to the gradient in Eq. (4), i.e.,*

$$\left\| \widehat{\frac{\partial \mathcal{L}}{\partial \boldsymbol{\theta}}} - \frac{\partial \mathcal{L}}{\partial \boldsymbol{\theta}} \right\| \leq \epsilon, \quad almost\ surely, \quad (19)$$

*then using Eq. (5) as a stochastic first-order oracle with a step size of $\eta_n = \mathcal{O}(1/\sqrt{n})$ to update $\boldsymbol{\theta}$ with gradient descent, it follows after $N$ iterations that*

$$\mathbb{E} \left[ \frac{\sum_{n=1}^N \eta_n \|\nabla \mathcal{R}(\boldsymbol{\theta}_n)\|^2}{\sum_{n=1}^N \eta_n} \right] \leq \mathcal{O} \left( \epsilon + \frac{\log N}{\sqrt{N}} \right). \quad (20)$$

**Remark 2.** Consider the condition in (19):

$$\left\| \widehat{\frac{\partial \mathcal{L}}{\partial \boldsymbol{\theta}}} - \frac{\partial \mathcal{L}}{\partial \boldsymbol{\theta}} \right\| \leq \left\| \boldsymbol{A} - \frac{\partial \mathcal{F}}{\partial \boldsymbol{\theta}} \left( \boldsymbol{I} - \frac{\partial \mathcal{F}}{\partial \boldsymbol{h}} \right)^{-1} \right\| \left\| \frac{\partial \mathcal{L}}{\partial \boldsymbol{h}} \right\|. \quad (21)$$

Suppose the gradient $\partial \mathcal{L} / \partial \boldsymbol{h}$ is almost surely bounded. By Theorem 2, the condition in (19) can be guaranteed as long as a sufficiently large $k$ is selected.

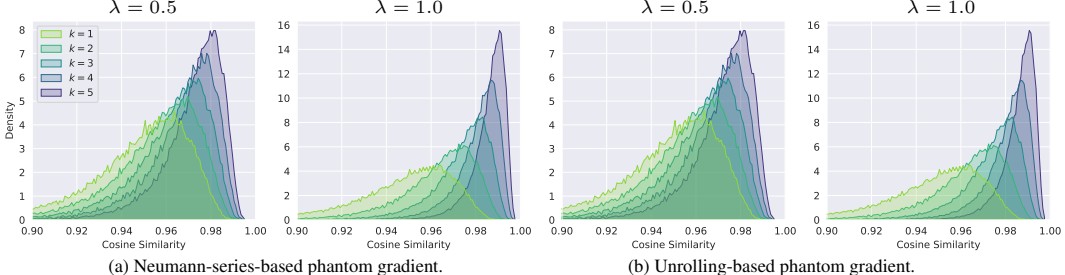

Figure 1: Cosine similarity between the phantom gradient and the exact gradient in the synthetic setting.

## 3 Experiments

In this section, we aim to answer the following questions via empirical results: (1) How precise is the phantom gradient? (2) What is the difference between the unrolling-based and the Neumann-series-based phantom gradients? (3) How is the phantom gradient influenced by the hyperparameters $k$ and $\lambda$? (4) How about the computational cost of the phantom gradient compared with implicit differentiation? (5) Can the phantom gradient work at large-scale settings for various tasks?

We have provided some theoretical analysis and intuitions to (1), (2), and (3) in Sec. 2.4. Now we answer (1) and (2) and demonstrate the performance curves under different hyperparameters $k$ and $\lambda$ on CIFAR-10 [31]. Besides, we also study other factors that have potential influences on the training process of the state-of-the-art implicit models [2, 3] like pretraining. For (4) and (5), we conduct experiments on large-scale datasets to highlight the ultra-fast speed and competitive performances, including image classification on ImageNet [10] and language modeling on Wikitext-103 [13].

We start by introducing two experiment settings. We adopt a single-layer neural network with spectral normalization [32] as the function $\mathcal{F}$ and fixed-point iterations as the equilibrium solver, which is the *synthetic setting*. Moreover, on the CIFAR-10 dataset, we use the MDEQ-Tiny model [3] (170K parameters) as the backbone model, denoted as the *ablation setting*. Additional implementation details and experimental results are presented in Appendix D[2].

**Precision of the Phantom Gradient.** The precision of the phantom gradient is measured by its angle against the exact gradient, indicated by the cosine similarity between the two. We discuss its precision in both the synthetic setting and the ablation setting. The former is under the static and randomly generated weights, while the latter provides characterization of the training dynamics.

In the synthetic setting, the function $\mathcal{F}$ is restricted to be a contraction mapping. Specifically, we directly set the Lipschitz constant of $\mathcal{F}$ as $L_{\boldsymbol{h}} = 0.9$, and use 100 fixed-point iterations to solve the root $\boldsymbol{h}^*$ of Eq. (1) until the relative error satisfies $\|\boldsymbol{h} - \mathcal{F}(\boldsymbol{h}, \boldsymbol{z})\|/\|\boldsymbol{h}\| < 10^{-5}$. Here, the exact gradient is estimated by backpropagation through the fixed-point iterations, and cross-validated by implicit differentiation solved with 20 iterations of the Broyden's method [16]. In our experiment, the cosine similarity between these two gradient estimates consistently succeeds 0.9999, indicating the gradient estimate is quite accurate when the relative error of forward solver is minor. The cosine similarity between phantom gradients and exact gradients is shown in Fig. 1. It shows that the cosine similarity tends to increase as $k$ grows and that a small $\lambda$ tends to slow down the convergence of the phantom gradient, allowing it to explore in a wider range regarding the angle against the exact gradient.

In the ablation setting, the precision of the phantom gradient during the training process is shown in Fig. 2. The model is trained by implicit differentiation under the official schedule[3]. It shows that the phantom gradient still provides an ascent direction in the real training process, as indicated by the considerable cosine similarity against the exact gradient. Interestingly, the cosine similarity slightly decays as the training progresses, which implies a possibility to construct an adaptive gradient solver for implicit models.

---

[2]All training sources of this work are available at https://github.com/Gsunshine/phantom_grad.
[3]Code available at https://github.com/locuslab/mdeq.

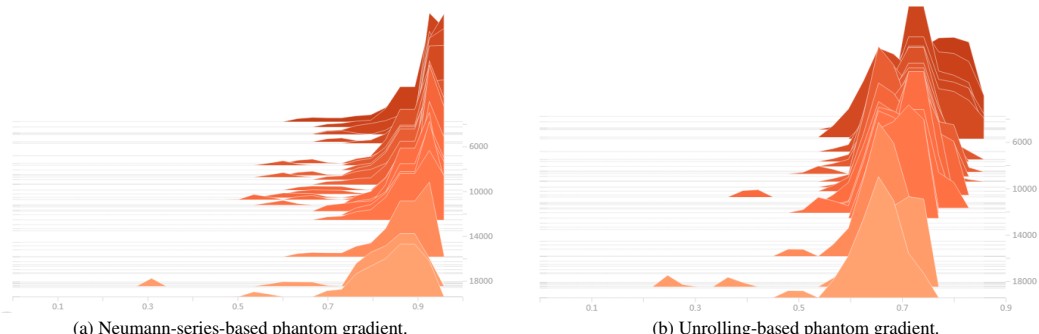

(a) Neumann-series-based phantom gradient.          (b) Unrolling-based phantom gradient.

Figure 2: Cosine similarity between the phantom gradient and the exact gradient in the real scenario. The horizontal axis corresponds to the cosine similarity, and the vertical axis to the training step.

**To Pretrain, or not to Pretrain?** To better understand the components of the implicit models' training schedule, we first illustrate a detailed ablation study of the baseline model in the ablation setting. The average accuracy with standard deviation is reported in Tab. 1.

The MDEQ model employs a pretraining stage in which the model $\mathcal{F}$ is unrolled as a recurrent network. We study the impact of the pretraining stage, the Dropout [33] operation, and the optimizer separately. It can be seen that the unrolled pretraining stabilizes the training of the MDEQ model. Removing the pretraining stage leads to a severe performance drop and apparent training instability among different trials because the solver cannot obtain an accurate fixed point $h^*$ when the model is not adequately trained. This ablation study also suggests that the MDEQ model is a strong baseline for our method to compare with.

Table 1: Ablation settings on CIFAR-10.

| Method | Acc. (%) |
|---|---|
| Implicit Differentiation | $85.0 \pm 0.2$ |
| w/o Pretraining | $82.3 \pm 1.3$ |
| w/o Dropout | $83.7 \pm 0.1$ |
| Adam $\rightarrow$ SGD | $84.5 \pm 0.3$ |
| SGD w/o Pretraining | $82.9 \pm 1.5$ |
| UPG ($\boldsymbol{A}_{5,0.5}$, w/o Dropout) | $\mathbf{85.8} \pm 0.5$ |
| NPG ($\boldsymbol{A}_{5,0.5}$, w/o Dropout) | $85.6 \pm 0.5$ |
| UPG ($\boldsymbol{A}_{9,0.5}$, w/ Dropout) | $\mathbf{86.1} \pm 0.5$ |

However, pretraining is not always indispensable for training implicit models. It introduces an extra hyperparameter, *i.e.,* how many steps should be involved in the pretraining stage. Next, we discuss how the UPG could circumvent this issue.

**Trade-offs between Unrolling and Neumann.** For an exact fixed point $h^*$, *i.e.,* $h^* = \mathcal{F}(h^*, z)$, there is no difference between UPG and NPG. However, when the numerical error exists in solving $h^*$, *i.e.,* $\|h^* - \mathcal{F}(h^*, z)\| > 0$, these two instatiations of the phantom gradient can behave differently.

We note that a particular benefit of the UPG is its ability to automatically switch between the pretraining and training stages for implicit models. When the model is not sufficiently trained and the solver converges poorly (see [3]), the UPG defines a forward computation graph that is essentially equivalent to a shallow weight-tied network to refine the coarse equilibrium states. In this stage, the phantom gradient serves as a backpropagation through time (BPTT) algorithm and hence behaves as in the pretraining stage. Then, as training progresses, the

Table 2: Complexity comparison. Mem means the memory cost, and $K$ and $k$ denote the solver's steps and the unrolling/Neumann steps, respectively. Here, $K \gg k \approx 1$.

| Method | Time | Mem | Peak Mem |
|---|---|---|---|
| Implicit | $\mathcal{O}(K)$ | $\mathcal{O}(1)$ | $\mathcal{O}(k)$ |
| UPG | $\mathcal{O}(k)$ | $\mathcal{O}(k)$ | $\mathcal{O}(k)$ |
| NPG | $\mathcal{O}(k)$ | $\mathcal{O}(1)$ | $\mathcal{O}(1)$ |

solver becomes more stable and converges to the fixed point $h^*$ better. This makes the UPG behave more like the NPG. Therefore, the unrolled pretraining is gradually transited into the regular training phase based on implicit differentiation, and the hyperparameter tuning of pretraining steps can be waived. We argue that such an ability to adaptively switch training stages is benign to the implicit models' training protocol, which is also supported by the performance gain in Tab. 1.

Although the UPG requires higher memory overhead than implicit differentiation or the NPG, it does not surpass the peak memory usage in the entire training protocol by implicit differentiation

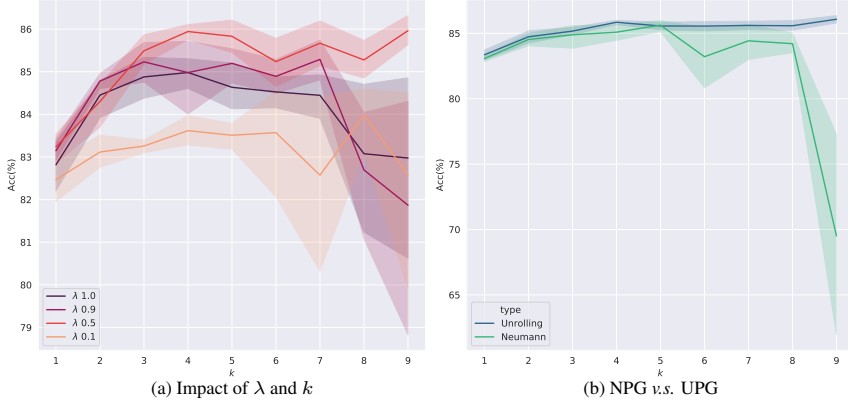

(a) Impact of $\lambda$ and $k$          (b) NPG $v.s.$ UPG

Figure 3: Ablation studies on (a) the hyperparameters $\lambda$ and $k$, and (b) two forms of phantom gradient.

due to the pretraining stage. In the ablation setting, the MDEQ model employs a 10-layer unrolling for pretraining, which actually consumes double memory compared with a 5-step unrolling scheme, *e.g.*, $\boldsymbol{A}_{5,0.5}$ in Tab. 1. In Tab. 2, we also demonstrate the time and memory complexity for implicit differentiation and the two forms of phantom gradient.

In addition, leaving the context of approximate and exact gradient aside, we also develop insights into understanding the subtle differences between UPG and NPG in terms of the state-dependent and state-free gradient. Actually, the UPG is state-dependent, which means it corresponds to the "exact" gradient of a computational sub-graph. Both the NPG and the gradient solved by implicit differentiation, however, do not exactly match the gradient of any forward computation graph unless the numerical error is entirely eliminated in both the forward and the backward passes for implicit differentiation or the one-step gradient [34, 21, 30, 22] is used for the NPG, *i.e.*, $k = 1$. Interestingly, we observe that models trained on the state-dependent gradient demonstrate an additional training stability regarding the Jacobian spectral radius, compared with those trained on the state-free counterpart. We empirically note that it can be seen as certain form of implicit Jacobian regularization for implicit models as a supplement to the explicit counterpart [17], indicated by the stable estimated Jacobian spectral radius during training, *i.e.*, $\rho(\partial \mathcal{F}_\lambda / \partial \boldsymbol{h}) \approx 1$.

The experimental results also cohere with our insight. The performance curves in Fig. 3 demonstrate the influence of $\lambda$ and $k$ and further validate that the UPG is more robust to a wide range of steps $k$ than the NPG. In fact, when the Jacobian spectral radius $\rho(\partial \mathcal{F} / \partial \boldsymbol{h})$ increases freely without proper regularization, the exploding high-order terms in the NPG could exert a negative impact on the overall direction of the phantom gradient, leading to performance degradation when a large $k$ is selected (see Fig. 3(b)). We also observe a surging of the estimated Jacobian spectral radius as well as the gradient explosion issue in the NPG experiments. On the contrary, the UPG can circumvent these problems thanks to its implicit Jacobian regularization.

Table 3: Experiments using DEQ [2] and MDEQ [3] on vision and language tasks. Metrics stand for accuracy(%)↑ for image classification on CIFAR-10 and ImageNet, and perplexity↓ for language modeling on Wikitext-103. JR stands for Jacobian Regularization [17]. † indicates additional steps in the forward equilibrium solver.

| Datasets | Model | Method | Params | Metrics | Speed |
|---|---|---|---|---|---|
| CIFAR-10 | MDEQ | Implicit | 10M | $93.8 \pm 0.17$ | $1.0\times$ |
| CIFAR-10 | MDEQ | UPG $\boldsymbol{A}_{5,0.5}$ | 10M | $95.0 \pm 0.16$ | $1.4\times$ |
| ImageNet | MDEQ | Implicit | 18M | 75.3 | $1.0\times$ |
| ImageNet | MDEQ | UPG $\boldsymbol{A}_{5,0.6}$ | 18M | 75.7 | $1.7\times$ |
| Wikitext-103 | DEQ (PostLN) | Implicit | 98M | 24.0 | $1.0\times$ |
| Wikitext-103 | DEQ (PostLN) | UPG $\boldsymbol{A}_{5,0.8}$ | 98M | 25.7 | $1.7\times$ |
| Wikitext-103 | DEQ (PreLN) | JR + Implicit | 98M | 24.5 | $1.7\times$ |
| Wikitext-103 | DEQ (PreLN) | JR + UPG $\boldsymbol{A}_{5,0.8}$ | 98M | 24.4 | $2.2\times$ |
| Wikitext-103 | DEQ (PreLN) | JR + UPG $\boldsymbol{A}_{5,0.8}$ | 98M | $24.0^\dagger$ | $1.7\times$ |

**Phantom Gradient at Scale.** We conduct large-scale experiments to verify the advantages of the phantom gradient on vision, graph, and language benchmarks. We adopt the UPG in the large-scale experiments. The results are illustrated in Tab. 3 and Tab. 4. Our method matches or surpasses the implicit differentiation training protocol on the state-of-the-art implicit models with a visible reduction on the training time. When only considering the backward pass, the acceleration for MDEQ can be remarkably $12\times$ on ImageNet classification.

Table 4: Experiments using IGNN [4] on graph tasks. Metrics stand for accuracy(%)↑ for graph classification on COX2 and PROTEINS, Micro-F1(%)↑ for node classification on PPI.

| Datasets | Model | Method | Params | Metrics (%) |
|---|---|---|---|---|
| COX2 | IGNN | Implicit | 38K | $84.1 \pm 2.9$ |
| COX2 | IGNN | UPG $\boldsymbol{A}_{5,0.5}$ | 38K | $83.9 \pm 3.0$ |
| COX2 | IGNN | UPG $\boldsymbol{A}_{5,0.8}$ | 38K | $83.9 \pm 2.7$ |
| COX2 | IGNN | UPG $\boldsymbol{A}_{5,1.0}$ | 38K | $83.0 \pm 2.9$ |
| PROTEINS | IGNN | Implicit | 34K | $78.6 \pm 4.1$ |
| PROTEINS | IGNN | UPG $\boldsymbol{A}_{5,0.5}$ | 34K | $78.4 \pm 4.2$ |
| PROTEINS | IGNN | UPG $\boldsymbol{A}_{5,0.8}$ | 34K | $78.6 \pm 4.2$ |
| PROTEINS | IGNN | UPG $\boldsymbol{A}_{5,1.0}$ | 34K | $78.8 \pm 4.2$ |
| PPI | IGNN | Implicit | 4.7M | 97.6 |
| PPI | IGNN | UPG $\boldsymbol{A}_{5,0.5}$ | 4.7M | 98.2 |
| PPI | IGNN | UPG $\boldsymbol{A}_{5,0.8}$ | 4.7M | 97.4 |
| PPI | IGNN | UPG $\boldsymbol{A}_{5,1.0}$ | 4.7M | 96.2 |

## 4  Related Work

**Implicit Models.** Implicit models generalize the recursive forward/backward rules of neural networks and characterize their internal mechanism by some pre-specified dynamics. Based on the dynamics, the implicit mechanisms can be broadly categorized into three classes: ODE-based [1, 7], root-solving-based [2, 3, 4, 5, 11], and optimization-based [35, 36, 37] implicit models.

The ODE-based implicit models [1, 7] treat the iterative update rules of residual networks as the Euler discretization of an ODE, which could be solved by any black-box ODE solver. The gradient of the differential equation is calculated using the *adjoint method* [38], in which the adjoint state is obtained by solving another ODE. The root-solving-based implicit models [2, 5, 3, 4, 6, 11, 22] characterize layers of neural networks by solving fixed-point equations. The equations are solved by either the black-box root-finding solver [2, 3] or the fixed-point iteration [4, 22]. The optimization-based implicit models [35, 36, 37, 39, 34, 21, 40, 41] leverage the optimization programs as layers of neural networks. Previous works have studied differentiable layers of quadratic programming [35], submodular optimization [36], maximum satisfiability (MAXSAT) problems [37], and structured decomposition [21]. As for the backward passes, implicit differentiation is applied to the problem-defining equations of the root-solving-based models [2, 3] or the KKT conditions of the optimization-based models [35]. As such, the gradient can be obtained from solving the backward linear system.

In this work, we focus on the root-solving-based implicit models. Theoretical works towards root-solving-based implicit models include the well-posedness [5, 4], monotone operators [11], global convergence [42, 41], and Lipschitz analysis [15]. We look into the theoretical aspect of the gradient-based algorithm in training implicit models and the efficient practice guidance. With these considerations, we show that implicit models of the same architecture could enjoy faster training speed and strong generalization in practical applications by using the phantom gradient.

**Non-End-to-End Optimization in Deep Learning.** Non-end-to-end optimization aims to replace the standard gradient-based training of deep architectures with modular or weakly modular training without the entire forward and backward passes. Currently, there are mainly three research directions in this field, namely, the auxiliary variable methods [43, 44, 45, 46, 47, 48, 49], target propagation [50, 51, 52], and synthetic gradient [24, 53, 54]. The auxiliary variable methods [43, 44, 45, 46, 47, 48, 49] formulate the optimization of neural networks as constrained optimization problems, in which the layer-wise activations are considered as trainable auxiliary variables. Then, the equality constraints are relaxed as penalty terms added to the objectives so that the parameters and auxiliary variables can be divided into blocks and thus optimized in parallel. The target propagation method [50, 51, 52] trains each module by having its activations regress to the pre-assigned targets, which are

propagated backwards from the downstream modules. Specifically, the auto-encoder architecture is used to reconstruct targets at each layer. Finally, the synthetic gradient method [24, 53, 54] estimates the local gradient of neural networks using auxiliary models, and employ the synthetic gradient in place of the exact gradient to perform parameter update. In this way, the forward and backward passes are decoupled and can be executed in an asynchronous manner.

Our work is in line with the non-end-to-end optimization research since we also aims to decouple the forward and backward passes of neural networks. However, we show that finding a reasonable "target" or a precise gradient estimate is not always necessary in training deep architectures. Our paper paves a path that an inexact but well-conditioned gradient estimate can contribute to both fast training and competitive generalization of implicit models.

**Differentiation through Implicit Dynamics.**  Differentiation through certain implicit dynamics is an important aspect in a wide range of research fields, including bilevel optimization [19, 9], meta-learning [18, 55, 30], and sensitivity analysis [56]. Since the gradient usually cannot be computed analytically, researchers have to implicitly differentiate the dynamics at the converged point. The formula of the gradient typically contains a term of Jacobian-inverse (or Hessian-inverse), which is computationally prohibitive for large-scale models. (See Eq. (2) in our case.) Herein, several techniques have been developed to approximate the matrix inverse in the previous literature.

An intuitive solution is to differentiate through the unrolled steps of a numerical solver of the dynamics [57, 58, 8]. In particular, if a single step is unrolled, it reduces to the well-known *one-step gradient* [59, 18, 60, 34, 30, 21, 22], in which the inverse of Jacobian/Hessian is simply approximated by an identity matrix. On the contrary, unrolling a small number of steps may induce a bias [9], while the memory and computational cost grows linearly as the number of unrolled steps increases. Towards this issue, Shaban *et al.* [19] propose to truncate the long-term dependencies and differentiate through only the last $L$ steps. In fact, if the dynamics have converged to a stationary point, the finite-term truncation in Shaban *et al.* [19] is exactly the Neumann approximation of the Jacobian-inverse with the first $L$ terms. Based on this, Lorraine *et al.* [9] directly use the truncated Neumann series as an approximation of the Jacobian-inverse. Besides the unrolling-based methods, optimization-based approaches [61, 55] have been studied in this field as well. Since the Jacobian-inverse-vector product can be viewed as solution of a linear system, algorithms like the conjugate gradient method can be used to solve it.

## 5   Limitation and Future Work

The main limitation of this work lies in the hyperparameter tuning of the phantom gradient, especially for the damping factor $\lambda$, which directly controls the gradient's precision and conditioning, the implicit Jacobian regularization for UPG, the stability for NPG, and the final generalization behaviors. However, it has not been a hindrance to the application of phantom gradients in training implicit models as one can tune the hyperparameter according to the validation loss in the early training stage.

Regarding future works, we would like to highlight the following aspects: (1) eliminating the bias of the current phantom gradient, (2) constructing an adaptive gradient solver for implicit models, (3) analyzing the damping factor to provide practical guidance, (4) investigating the implicit Jacobian regularization, and (5) understanding how different noises in the gradients can impact the training of implicit models under different loss landscapes.

## 6   Conclusion

In this work, we explore the possibility of training implicit models via the efficient approximate phantom gradient. We systematically analyze the general condition of a gradient estimate so that the implicit model can be guaranteed to converge to an approximate stationary point of the loss function. Specifically, we give a sufficient condition under which a first-order oracle could always find an ascent direction of the loss landscape in the training process. Moreover, we introduce two instantiations of the proposed phantom gradient, based on either the damped fixed-point unrolling or the Neumann series. The proposed method shows a $1.4 \sim 1.7\times$ acceleration with comparable or better performances on large-scale benchmarks. Overall, this paper provides a practical perspective on training implicit models with theoretical guarantees.

**Acknowledgments**

Zhouchen Lin was supported by the NSF China (No.s 61625301 and 61731018), NSFC Tianyuan Fund for Mathematics (No. 12026606) and Project 2020BD006 supported by PKU-Baidu Fund. Yisen Wang was partially supported by the National Natural Science Foundation of China under Grant 62006153, and Project 2020BD006 supported by PKU-Baidu Fund. Shaojie Bai was sponsored by a grant from the Bosch Center for Artificial Intelligence.

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
