# OpenReview forum: "On Training Implicit Models"
_NeurIPS.cc/2021/Conference — NeurIPS 2021 Poster_

### Official Review · Reviewer_kurx · 2021-07-15

**Rating:** 6
**Confidence:** 3

**Summary:**

This paper proposes a new method to estimate gradient for implicit models, which empirically accelerates  the training of implicit models.

**Ethical Concerns:**

No ethical concern is identified.

**Main Review:**

Overall this is an interesting paper. The major contributions are:

1. This paper proposes a new method to estimate gradient in training implicit models, where the idea is quite straight-forward and easy to follow;
2. Some analyses are provided on how well the approximation could be, including some theoretical results and their intuitive explanations;
3. Experimental results on CIFAR-10 and ImageNet are also provided, in which the proposed method shows its effectiveness in terms of accuracy and efficiency.

Below are some other comments and questions:

1. There are three major theorems in the paper. The first one relies on the condition  based on singular values and the last one relies on the condition of good approximation. As mentioned in the paper, the condition for the first theorem is related to  NTK. Consequently, when the model is 'sufficiently wide', the condition can be easily satisfied. However, Theorem 3.1 & 3.2 in [Arora et al. 2019] suggest that this 'sufficiently wide' requirement could be rather impractical, as the required width  is much wider than those commonly used models in practice. Thus, the conditions for the first and third theorems are somewhat strong. Maybe the authors can consider adding an experiment to study the impact of different model designs.
2. It would strengthen the paper if it compares more related works in experiments and reports all the results with mean (standard deviation);
3. In the experiments, the proposed method leads to worse results sometime. It would be interesting to know whether this is due to the approximation quality or something else. Hope the authors can provide some detailed discussions on this.


**Time Spent Reviewing:**

6 hours.

---

> ### Author Response · Authors · 2021-08-10
> **Response to Reviewer kurx**
>
> Thanks for the comments! You have confirmed the novelty and clarity of the method as well as the experimental results. Concerning other questions, we are happy to share the responses below.
>
> > The conditions for the first and third theorems are somewhat strong due to the NTK.
>
> Firstly, we have to clarify that Remark 2 in our paper is just an illustrating example to explain the conditions in (6) and (8). (The singular values in (6) and the condition number in (8) are not intuitive at the first sight.) We do not intend to verify the condition in (6) or (8) with the NTK theory; instead, the condition in (6) is verified by Thm. 2 and the inequity in (19), whose assumption is just the contractive-ness of F (it will be relaxed to the spectral radius less than 1) and a sufficiently large $k$. Similarly, the condition (20) of Thm. 3 is verified by Thm. 2 and the inequity in (22). Moreover, Fig. 2 demonstrates that the cosine similarity is indeed positive when training with the phantom gradient in the real scenario, implying that the selected $k$ is actually sufficiently large so that the condition in (6) is satisfied.
>
> > It would strengthen the paper if it compares more related works in experiments and reports all the results with mean (standard deviation).
>
> The error bars are plotted in Fig. 3. In Tab. 1, we also report the performances from six independent runs in the form of _mean(best)_. We will also include the standard derivation in the final version. Besides, in Tab. 3, the performance statistics of MDEQ (10M) on CIFAR-10 is also averaged from three independent runs ($95.0 \pm 0.16$% from UPG v.s. $93.8 \pm 0.17$% from implicit differentiation, 10M parameters).
>
> As for comparing with related work, we will add more baselines as previous work [1]. Concerning the transferability to different implicit models and tasks, we conduct some preliminary experiments on graph neural networks using IGNN [2], achieving $83.9 \pm 3.0$ acc(%) for COX2 graph classification compared with $84.1 \pm 2.9$ using implicit differentiation, $78.6 \pm 4.1$ acc(%) for PROTEINS graph classification compared with $78.6 \pm 4.0$ using implicit differentiation. The results are therefore consistent with our findings on vision tasks.
>
> [1] Bai, Shaojie, Vladlen Koltun, and J. Zico Kolter. "Multiscale Deep Equilibrium Models." Advances in Neural Information Processing Systems 33 (2020).
>
> [2] Gu, Fangda, et al. "Implicit Graph Neural Networks." Advances in Neural Information Processing Systems 33 (2020): 11984-11995.
>
> > In the experiments, the proposed method leads to worse results sometime. It would be interesting to know whether this is due to the approximation quality or something else. Hope the authors can provide some detailed discussions on this.
>
> Empirically, in most of the cases, we observe that the proposed method achieves performances on par with the baseline methods. In fact, since the submission we have been able to obtain results on ImageNet classfication that surpass the baseline by tuning the hyperparameters and pretraining (75.0 -> 75.7). The given approximation quality for MDEQ therefore is good enough for training the model to the state-of-the-art performances.
>
> More broadly, regarding the potential failure modes and the underlying mechanism, we hypothesize that the loss landscape of implicit models (which consists of the model architecture, internal attributes of the dataset, regularizations, etc.), combined with the choice of optimizers and the noise distribution, jointly determine the final generalization performance. Therefore, perhaps it is more accurate to consider the various attributes of approximation because both the noise scale (reflected by the cosine similarity) and the shape of noise can impact the dynamics (e.g., SGD is by itself also noisy/stochastic). This is a very good but deep question that requires more investigations in future research, as it may be a function of multiple factors at play here. We would like to thank the reviewer for raising this point and are happy to discuss this more in the revision.

---

### Official Review · Reviewer_hVtm · 2021-07-15

**Rating:** 7
**Confidence:** 4

**Summary:**

The paper presents a method called phantom gradient to increase the efficiency in training implicit deep models by approximating the exact gradient. The authors first exploit Neumann series for the damped inverse matrix in implicit gradient calculation. They then truncate the series to k terms to arrive at finite term approximation of the exact implicit gradient. They finally provide theoretical justification for when phantom gradients are descent directions for implicit models. The authors evaluate their methods by conducting experiments on thoroughly analyzing their descent direction and demonstrating acceleration over a current implicit training method.

**Limitations And Societal Impact:**

See main review.

**Main Review:**

[Strength] The paper has the following strength.
- The paper studies the very interesting and important problem of accelerating implicit model training. Which is also a difficult problem. The paper gives a novel insight that memorizing the first few steps of the forward fixed point iterations can help more efficient and stable gradient evaluation.
- The proposed method gives theoretical analysis of the method on the validity of phantom gradients and convergence analysis of the method.
- The method offers significant speedup in backward steps over the current implicit gradient method in model training and offer a moderate speedup overall.
- Thorough empirical analysis on the property of the model is well conducted in the paper, showing cleverly chosen parameters are key to the speedup in implicit training. It is also demonstrated that the method scales well to large experiments in computer vision.

[Weekness] The paper is well written but has a few glitches. I think the paper would be a good addition to NeurIPS if the authors can carefully address them.
- The notations of the paper are not clearly given. (e.g. The norm and inner product of Theorem 1). I sugguest they are operator norm and matrix inner product. Though careful inspection could infer the notations, a clearer definition can help make the paper more readable for general public.
- The paper should more adequately discuss related works such as (El Ghaoui, 2019; Revay, 2020; Gu, 2020) on theoretical contributions like well-posedness (refereed as well-conditioned in this paper) for implicit models and (Wang, 2020; Gu, 2020) on recent applications of implicit models on object detection and graph neural networks.
- I have seen a varient of Theorem 2 in Theorem 6.1 of (El Ghaoui, 2019). I think the condition can be further relaxed to something like $\partial F / \partial h*$ has spectual radius less than 1. Note that this is not the same as being contractive. I think the relation should not undermine the contribution of this paper though.
- The parameter choices of k and lambda are not well-connected with the theoretical analysis as in Theorem 1. It would help the consistency if the empirical choices of the parameters in accordance to the theory can be discussed in section 2.3. Also in section 2.3 (i) and (ii), the benefits of the better conditioning of A matrix is not well discussed. It only start to make sense to me after reading the experiment section for a while. I sugguest it helps the accuracy of the model although resulting in less accurate gradient.. Right?
- Figure 3 (b) could use other colors for UPG and NPG to avoid visual confusion.
- The paper could also include speed comparison on just the backward steps as I suggest it would show a much more visible improvement. Because the method does not touch the forward pass which takes roughly the same amount of time to do as the backward, the overall improvement cannot exceed approximately 2x, making it appear less impressive.
- The proposed method should be general to all implicit models. The authors currently fail to show this by only offering experiments regarding computer vision tasks. I think the impact from generalization capability of the method would be better demonstrated if the author can include experiments on tasks in the final version, for example on natural language processing (Bai, 2019) and/or graph neural network (Gu, 2020) experiments whose code is available online.
- I hope the authors be more generous in sharing the limitations of the method as suggested by NeurIPS 2021 board (see checklist 1b).

Ref:

Bai, S., Kolter, J. Z., & Koltun, V. (2019). Deep equilibrium models. Advances in Neural Information Processing Systems 32. https://papers.nips.cc/paper/2019/hash/01386bd6d8e091c2ab4c7c7de644d37b-Abstract.html

El Ghaoui, L., Gu, F., Travacca, B., Askari, A., & Tsai, A. Y. (2019). Implicit deep learning. arXiv preprint arXiv:1908.06315, 2. https://arxiv.org/abs/1908.06315

Revay, M., Wang, R., & Manchester, I. R. (2020). Lipschitz Bounded Equilibrium Networks. arXiv preprint arXiv:2010.01732. https://arxiv.org/abs/2010.01732

Gu, F., Chang, H., Zhu, W., Sojoudi, S., El Ghaoui, L. (2020). Implicit graph neural networks. Advances in Neural Information Processing Systems 33, 11984--11995. https://papers.nips.cc/paper/2020/hash/8b5c8441a8ff8e151b191c53c1842a38-Abstract.html

Wang, T., Zhang, X., & Sun, J. (2020). Implicit feature pyramid network for object detection. arXiv preprint arXiv:2012.13563. https://arxiv.org/abs/2012.13563


---------------------------------------
Post Rebuttal:
I appreciate the authors' detailed responses to my questions. I am glad that the authors find my comments useful and will include them in the final draft. I have no further concerns about the paper being accepted by NeurIPS. Thus, I will update my score to 7.

**Time Spent Reviewing:**

8

---

> ### Author Response · Authors · 2021-08-10
> **Response to Rviewer hVtm**
>
> We are very pleased to receive the thorough comments, affirming the significance of the research problem, the novelty of our idea, sound theories and noteworthy empirical accelerations. We would like to make best efforts to solve the remaining concerns.
>
> > The notations of the paper are not clearly given.
> >
> > Figure 3 (b) could use other colors for UPG and NPG to avoid visual confusion.
>
> Thanks for the comments! The clarity of the paper will be improved in the revision. We plan to add a table of notions and make the plots clearer.
>
> > The paper should more adequately discuss related works such as (El Ghaoui, 2019; Revay, 2020; Gu, 2020) on theoretical contributions like well-posedness (refereed as well-conditioned in this paper) for implicit models and (Wang, 2020; Gu, 2020) on recent applications of implicit models on object detection and graph neural networks.
>
> Thanks for introducing the valuable literature! The theoretical analysis and applications of implicit models will be elaborated in the "Related Work" section. We will also highlight the discussion on well-posedness and its connection to our method.
>
> > The parameter choices of k and lambda are not well-connected with the theoretical analysis as in Theorem 1. It would help the consistency if the empirical choices of the parameters in accordance to the theory can be discussed in section 2.3. Also in section 2.3 (i) and (ii), the benefits of the better conditioning of A matrix is not well discussed. It only start to make sense to me after reading the experiment section for a while. I sugguest it helps the accuracy of the model although resulting in less accurate gradient.. Right?
>
> Yes. We will adjust the discussion in Section 2.3 to make the logic smoother. The choice of $k$ and $\lambda$ implies a trade-off between the accuracy of gradients and conditioning, reflecting in the final performance. It also possibly relates to the very recently discussed topics regarding the stability of implicit models and Jacobian regularization [1]. We think that it would be very interesting to understand the underlying mechanism from this perspective as one of the future directions.
>
> > Relax the condition in Theorem 2.
>
> Thanks for the suggestion. We will rewrite the condition via spectral radius rather than being contractive (indeed helpful). The relation will be mentioned in the "Related Work" section as well.
>
> > The paper could also include speed comparison on just the backward steps.
>
> Good suggestion! We test the speed according to the comment. The acceleration can be **remarkable $12\times$** for MDEQ on ImageNet classification when only taking the backward pass into account. We would like to highlight the accelerations!
>
> > The proposed method should be general to all implicit models.
>
> Our preliminary experiment does show that the proposed lightweight phantom gradient offers promising results in other tasks, like DEQ-Transformers for WikiText-103 language modeling, where we can train the model with 1.7x acceleration to a competitive level (e.g., currently 25.2 ppl). This is therefore consistent with our findings on vision tasks. Meanwhile, we also notice that the optimal choice of $\lambda$ and $k$ may vary with the specific model architecture, which potentially suggests a combination of the phantom gradient and other recently proposed techniques like Jacobian regularization [1] or provably convergent implicit networks [2,3].
>
> We try the DEQ-Transformers' setting under Jacobian regularization [1] and observe that the phantom gradient could produce a similar val ppl curve compared with implicit differentiation while obtaining a 2.2x acceleration. (Jacobian regularization also reduces the forward computations). Regarding the IGNN setting [2], the phantom gradient achieves 98.2 Micro F1(%) performance for PPI node classification compared with 97.6 using implicit differentiation, $83.9 \pm 3.0$ acc(%) for COX2 graph classification compared with $84.1 \pm 2.9$ using implicit differentiation, $78.6 \pm 4.1$ acc(%) for PROTEINS graph classification compared with $78.6 \pm 4.0$ using implicit differentiation. These primary results validate that the phantom gradient can be a potentially efficient alternative to implicit differentiation in these tasks.
>
> We are happy to include these results and discussions further in the revision.
>
> [1] Bai, Shaojie, Vladlen Koltun, and Zico Kolter. "Stabilizing Equilibrium Models by Jacobian Regularization." International Conference on Machine Learning. PMLR, 2021.
>
> [2] Gu, Fangda, et al. "Implicit Graph Neural Networks." Advances in Neural Information Processing Systems 33 (2020): 11984-11995.
>
> [3] Revay, Max, Ruigang Wang, and Ian R. Manchester. "Lipschitz Bounded Equilibrium Networks." arXiv preprint arXiv:2010.01732 (2020).
>
> > The limitations of the method.
>
> Regarding the limitations and future works, we would like to add this section to the final version, including hyperparameter tuning, the connection with the very recently proposed stability of implicit models [1], and understanding how different noises in the gradients can impact the optimization and generalization of implicit models.
>
> The main scope of our paper is that a cheap gradient estimate can help train a black-box implicit layer and has been carefully demonstrated in the paper, while these limitations could still inspire future research.
>
> [1] Bai, Shaojie, Vladlen Koltun, and Zico Kolter. "Stabilizing Equilibrium Models by Jacobian Regularization." International Conference on Machine Learning. PMLR, 2021.
>
> ----
>
> Thank you again for perusing this work and providing constructive comments! We are happy to add the revision suggested by the comments to the final version.

---

### Official Review · Reviewer_FWcz · 2021-07-16

**Rating:** 7
**Confidence:** 4

**Summary:**

The paper proposes an approximate gradient, called phantom gradient, for training implicit models. They highlight that the method reduces the cost of gradient computation and it provides a descent direction suitable to train models. Their gradients are based on two methods, fixed-point unrolling and the Neumann series. The paper proves that under some network conditions on the eigenvalues of the derivative of the implicit module, the phantom gradient has a descent direction positively corelated to the exact gradient. The paper also provide almost surely convergence of the phantom gradient to the exact gradient under some conditions.

**Ethical Concerns:**

No.

**Limitations And Societal Impact:**

Limitations is not included. A need for tuning lambda can be one limitation of the gradient.

I suggest to include some societal impact related to the computational cost of training implicit models and their robustness.

**Main Review:**

The paper is nicely written and is well-organized. The paper explains well how their method differs from prior works. Although the work is incremental, the approximate gradient due to its reduced cost computation is of interest. Here are my detailed comments.

- The term (1 - lambda) h_t in (13) is introducing a momentum. Please discuss the results from the perspective of momentum.

- It would be nice to emphasize that when \lambda = 1, the proposed gradient is reduced to backprop through the unrolled iterations (e.g., Figure 1b right). All reported results are based on lambda = 0.5, and k = 5. a) the choice reason should be explained. b) the experiments should cover a range for lambda and k, specially the case lambda = 1 (backprop but not exact gradient). Indeed, their approach should be compared against the case lambda = 1 to highlight the advantages of the phantom gradient.


- At line 12 and 54, please be more specific that the speed up is compared to which methods (backprop, which implicit diff method)?

- It would be nice to move some of the questions asked on paragraph line 174 to the introduction or contribution.

- Although included in the supp, explain for the tables that A_{5,0.5} is for k = 5, and lambda = 0.5.

- Place the tables and figures close to their descriptions.

- It's nice that the code has been made available. I suggest to include a readme.

Minor comments:

- typo on supp line 20. It should be Remark.



----------------------------------------------------------------------------------------------------------------
Post response review
----------------------------------------------------------------------------------------------------------------
----------------------------------------------------------------------------------------------------------------

I have found the authors' response to the reviews satisfactory. I strongly recommend to add some of the discussions made here to either the paper or supplementary, specifically elaboration on NTK, hyperparameter tuning, result discussion.

Overall, given the modifications suggested by the reviewers, I have increased my score.

**Time Spent Reviewing:**

3

---

> ### Author Response · Authors · 2021-08-10
> **Response to Reviewer FWcz**
>
> Thanks for the detailed comments! You offer positive comments on the clarity, novelty, and especially the experimental accelerations. Regarding the concerns, responses are as follows.
>
> > The term (1 - lambda) h_t in (13) is introducing a momentum. Please discuss the results from the perspective of momentum.
>
> The damping factor $\lambda$ is a kind of momentum that introduces history information into the solver. Previous work [1,2,3] shows that the damping increases the radius of convergence and can adequately handle the hard case in the solving process. In this paper, the damping effect also plays a vital role for phantom gradients. It helps implicit models without provable convergence (like MDEQ) obtain smoother gradient estimates and become more stable.
>
> [1] Fang, Haw‐ren, and Yousef Saad. "Two classes of multisecant methods for nonlinear acceleration." Numerical Linear Algebra with Applications 16.3 (2009): 197-221.
>
> [2] Walker, Homer F., and Peng Ni. "Anderson acceleration for fixed-point iterations." SIAM Journal on Numerical Analysis 49.4 (2011): 1715-1735.
>
> [3] Evans, Claire, et al. "A proof that Anderson acceleration improves the convergence rate in linearly converging fixed-point methods (but not in those converging quadratically)." SIAM Journal on Numerical Analysis 58.1 (2020): 788-810.
>
> > The choice of hyperparamters $k$ and $\lambda$ and the comparison with $\lambda=1$.
>
> Fig. 3(a) offers an extensive ablation study on the impact of hyperparameters $\lambda$ and $k$, in which $\lambda=0.5$ shows favorable performance over $\lambda=1$. Empirically, we also observe that the selected hyperparameters can transfer well among different models and different datasets. Moreover, we do not exclude the case $\lambda=1$ from the phantom gradient because it can also produce a valid gradient estimate using our methods. We will highlight it in the final version.
>
> > At line 12 and 54, please be more specific that the speed up is compared to which methods (backprop, which implicit diff method)?
>
> The speed up is compared with the default backward pass of MDEQs using implicit differentiation (implemented via the Broyden method). The acceleration is computed using the same codebase. We will clarify it in the revision.
>
> > It would be nice to move some of the questions asked on paragraph line 174 to the introduction or contribution.
>
> Thanks for the suggestions! We note that the summary of experiments receives positive comments from different reviewers. We are pleased to improve the clarity further through this technique.
>
> > Although included in the supp, explain for the tables that A_{5,0.5} is for k = 5, and lambda = 0.5.
> > Place the tables and figures close to their descriptions.
> > Typo on supp line 20. It should be Remark.
>
> The writing issues will be fixed in the revision. Thanks!
>
> > It's nice that the code has been made available. I suggest to include a readme.
>
> Of course! We plan to release the training sources once the paper gets accepted. A detailed README will certainly be added to the repo, including both CIFAR and ImageNet experiments.
>
> ----
>
> Regarding the limitations and future works, we would like to add this section to the final version, including hyperparameter tuning, the connection with the very recently proposed stability of implicit models [1], and understanding how different noises in the gradients can impact the optimization and generalization of implicit models.
>
> We are also optimistic to the social impact of reducing the carbon footprint for training implicit models. :)
>
> [1] Bai, Shaojie, Vladlen Koltun, and Zico Kolter. "Stabilizing Equilibrium Models by Jacobian Regularization." International Conference on Machine Learning. PMLR, 2021.
>
> ----
>
> ## Reply to the response
>
> ----
>
> Thank you very much for your reply and comments to the rebuttal! We will take your suggestion and further refine its scientific value at the best. Thanks again for reviewing this paper!

---

### Official Review · Reviewer_f2Rd · 2021-07-16

**Rating:** 7
**Confidence:** 3

**Summary:**

This paper focuses on the problem efficiently training Deep Equilibrium Models (DEQs) by estimating the gradient to the "equilibirum layer." Usually, this gradient is computed using a black-box fixed point solver (generally Broyden's), which makes the backward pass expensive. This paper introduces a gradient estimate, which they term the _phantom gradient_. They show that the dot product between this gradient and the true gradient is greater than 0, which means it constitutes a descent direction in a first-order sense. They empirically verify that these estimates are good using CIFAR-10 and ImageNet, where they get performance remarkably close to the standard Multiscale DEQ [1] with a reasonable training speed improvement. They also show that the phantom gradient approaches the true gradient in terms of cosine similarity with increasing series elements $k$.

**Ethical Concerns:**

This paper does not have direct ethical issues.

**Limitations And Societal Impact:**

They do not discuss the limitations or social impact of their work. Given that this paper is rather abstract, dealing with a learning method rather than an application, I believe this to be justified. One suggestion for limitations, however, would be to try to study cases where their gradient estimator is unstable.

**Main Review:**

### Strengths

1. The paper is well-written and organized very well. Every step is motivated properly and the remarks provide good intuition for the theory.
2. The empirical experimentation is reasonably thorough, in particular lines (174-179) provide a good roadmap for section 3.
3. The speed improvement is significant. The results on ImageNet show that this algorithm scales well with minimal performance degradation (75.3 -> 75.0).
4. The convergence theory provided in section 2.4 along with the cosine alignment plots in Figure 1 are very convincing.
5. The relevant hyperparameters of the method are ablated appropriately (Figure 3).
6. I like the takeaway that precise gradient estimates are not always required, especially in cases where a layer is a black box. I think this will be of interest to the community.

### Weaknesses

1. Remark 2 seems incorrect. This holds in the linear model case, however for generic multilayer networks with ReLU or sigmoid non-linearity, the NTK is not a scalar network. See [2] for details. Perhaps there's some other effect keeping the Jacobian well-conditioned with large width however.
2. Does the Neumann Series for computing the phantom gradient always converge faster than Broyden's fixed point in the standard DEQ setup? It seems like this algorithm can be seen as trading one iterative solver for another.
3. There are no error bars for the small scale experiment. Is training with this gradient estimate stable in terms of performance?


Overall I think this method is quite interesting and will be of benefit to the larger NeurIPS community. My rating can be improved if the authors answer some of the questions I have posed, or clarify some misconceptions with their method.


### Questions to the Author

1. Is the Jacobian-inverse likely to be ill-conditioned in this setting, or is this mostly a motivation?
2. I would like to see a plot of the condition in Eq. 6, perhaps for a small dataset. Is this a reasonable assumption?
3. Figure 2 shows that cosine similarity with the exact gradient decreases with the training iterations. How does this affect training later on? Could the algorithm be improved by keeping $k$ low early in training and increasing it later?
4. Are there cases where Broyden converges faster than the Neumann series?
5. Do you find that these hyperparameters transfer well between tasks? Do you need to re-tune?


[1] Shaojie Bai, Vladlen Koltun, and J. Zico Kolter. Multiscale Deep Equilibrium Models. In Neural Information Processing Systems (NeurIPS), pages 5238–5250, 2020.
[2] Youngmin Cho, Lawrence Saul. Kernel Methods for Deep Learning. In Advances in Neural Information Processing Systems 22 (NIPS 2009)

**Time Spent Reviewing:**

4

---

> ### Author Response · Authors · 2021-08-10
> **Response to Reviewer  f2Rd**
>
> Thanks for the detailed review! We are glad to receive the kudos on the writing of this paper, the core idea itself, and the performances with accelerations. It is a great pleasure that our efforts (e.g., the roadmap) bring pleasant reading experiences to our reviewers.
>
> Regarding the weaknesses and questions, our responses are as follows.
>
> > Remark 2 seems incorrect. This holds in the linear model case, however for generic multilayer networks with ReLU or sigmoid non-linearity, the NTK is not a scalar network.
>
> We have to clarify that the neural tangent kernel refers to the one proposed in [1] instead of the kernel function in support vector machines (SVMs) or multilayer kernel machines (MKMs). As long as the neural network is constructed by stacking linear and nonlinear operators sequentially, the convergence in Eq. (9) in our paper holds. See Sec. 2 of [1] for fully-connected neural networks and Sec. 4 of [2] for convolutional neural networks.
>
> [1] Arthur Jacot, Franck Gabriel, and Clement Hongler. Neural Tangent Kernel: Convergence and Generalization in Neural Networks. In Neural Information Processing Systems (NeurIPS), pages 8571–8580, 2018.
>
> [2] Sanjeev Arora, Simon S Du, Wei Hu, Zhiyuan Li, Russ R Salakhutdinov, and Ruosong Wang. On Exact Computation with an Infinitely Wide Neural Net. In Neural Information Processing Systems (NeurIPS), pages 8139–8148, 2019.
>
> > Are there cases where Broyden converges faster than the Neumann series?  It seems like this algorithm can be seen as trading one iterative solver for another.
>
> Regarding convergence, Broyden's method can be faster than Neumann series in the well-conditioned cases (for exact convergence) because the former is a quasi-Newton method. However, in the context of approximate gradient, we believe that it is also interesting to discuss this comparison beyond the scope of only the convergence of gradient estimates.
>
> Specifically, as the reviewer mentioned, the core idea of this paper is that precise gradient estimates are not always required for training implicit models (or a black-box layer). To create valid approximate gradients (namely, phantom gradients), we first consider early stopping a solver (i.e., truncation), and then construct a proper "gradient solver" with theoretical guarantees. Therefore, performing such analysis in the context of approximate gradients entails considerations/criteria in more axes (than just convergence rate): e.g., monotonicity, stability, performance, etc., in this "inexact" setting. Therefore, our goal of analyzing whether inexact gradients **can be used to train** implicit models is only marginally impacted by the exact solver choice, and we are not trading anything per se here (e.g., our preliminary experiments suggest that highly inexact Broyden also works). In our analysis, the Neumann solver much more naturally aligns with the unrolling pretraining stage of the prior works [1]. Meanwhile, we provide the theoretical guarantee on the convergence when training using phantom gradients via the Neumann solver. Empirically, Figure 1 in the supplement also validates that the Neumann solver has better stability than the Broyden solver in the ill-conditioned cases (which is one of the motivations), a property that is generally appreciated by both exact and approximate gradients. We will clarify this further in the final version of this paper.
>
> [1] Bai, Shaojie, Vladlen Koltun, and J. Zico Kolter. "Multiscale Deep Equilibrium Models." Advances in Neural Information Processing Systems 33 (2020).
>
> > There are no error bars for the small scale experiment. Is training with this gradient estimate stable in terms of performance?
>
> The error bars are plotted in Fig. 3. In Tab. 1, we also report the performances from six independent runs in the form of _mean(best)_. We will also include the standard derivation in the final version. Besides, in Tab. 3, the performance statistics of MDEQ (10M) on CIFAR-10 is also averaged from three independent runs.
>
> Concerning stability, phantom gradients show similar training variance compared with implicit differentiation. It shows slightly larger variance than the baseline model in the tiny setting ($85.8 \pm 0.46$% v.s. $85.0 \pm 0.24$%, 170K parameters, 6 runs) and comparable variance for overparameterized models ($95.0 \pm 0.16$% v.s. $93.8 \pm 0.17$%, 10M parameters, 3 runs).
>
> > Is the Jacobian-inverse likely to be ill-conditioned in this setting, or is this mostly a motivation?
>
> We observe that the Jacobian occasionally becomes ill-conditioned when training with phantom gradients. As discussed, the proposed phantom gradient is more stable in ill-conditioned cases than the Broyden solver.
>
> The ill-conditionness of Jacobian is part of the motivation, but slightly subordinate to the expensive exact gradient in implicit differentiation. The ill-conditioned cases can be corner cases, while the expensive backward pass actually occurs in each optimization step.
>
> > I would like to see a plot of the condition in Eq. 6, perhaps for a small dataset. Is this a reasonable assumption?
>
> We verify the condition in (6) using the synthetic setting. We calculate the LHS and RHS using an increasing $k$ and find the condition satisfied when a large enough $k$ is selected because increasing $k$ can reduce the LHS while the RHS is a fixed number. We will add this plot in the final version.
>
> > Figure 2 shows that cosine similarity with the exact gradient decreases with the training iterations. How does this affect training later on? Could the algorithm be improved by keeping $k$ low early in training and increasing it later?
>
> Interesting question! We believe that the decrease of cosine similarity does not visibly impede training or harm generalization in the sense of average performance but may relate to the performance variance among different runs. The problem is that we cannot always obtain the exact gradient and compute the cosine similarity. So it is not easy to determine the proper time and range to change $k$. We try to increase $k$ from 5 to 6 at the 12K iterations under the ablation setting, obtaining $85.8 \pm 0.40$% compared with $85.8 \pm 0.46$% in the original fixed-$k$ setting. This topic can be further investigated in future research to see whether it is possible to construct an adaptive gradient solver for implicit models.
>
> > Are there cases where Broyden converges faster than the Neumann series?
>
> See answers above. :) Thanks!
>
> > Do you find that these hyperparameters transfer well between tasks? Do you need to re-tune?
>
> The hyperparameter $k$ transfers well. We recommend to re-tune the hyperparameter $\lambda$ for different tasks. The combination of $k=5$ and $\lambda=0.5$ is tested from the ablation setting and can obtain satisfactory performances using the large model on both CIFAR-10 and ImageNet datasets. Using $\lambda=0.6$ on ImageNet can improve the performance to 75.2%. Combining with the automatic pretraining for UPG, the performance can be further boosted to 75.4% for $\lambda=0.5$ and 75.7% for $\lambda=0.6$, even surpassing 75.3% using the exact gradient strategy on ImageNet.
>
> In general, the provided hyperparameters can produce acceptable results. Further tuning these hyperparameters can show additional but relatively marginal gains. The optimal $\lambda$ may range from 0.5 to 1.0, thus not raising the difficulty of tuning hyperparameters.
>
> ----
>
> Regarding the limitations and future works, we would like to add this section to the final version, including hyperparameter tuning, the connection with the very recently proposed stability of implicit models [1], and understanding how different noises in the gradients can impact the optimization and generalization of implicit models.
>
> The main scope of our paper is that a cheap gradient estimate can help train a black-box implicit layer and has been carefully demonstrated in the paper, while these limitations could still inspire future research.
>
> [1] Bai, Shaojie, Vladlen Koltun, and Zico Kolter. "Stabilizing Equilibrium Models by Jacobian Regularization." International Conference on Machine Learning. PMLR, 2021.

---

> > ### Comment · Reviewer_f2Rd · 2021-08-31
> > **Response to Rebuttal**
> >
> > Thank you to the authors for providing a thoughtful rebuttal which answered most of my concerns. After reviewing the other reviews and author responses, I recommend acceptance with a score of 7 with the changes the authors will make.
> >
> > To the authors, I believe your understanding of the NTK is incorrect. I was not referring to kernels from SVMs in particular, but a derivation of the ReLU kernel in NNGPs which is used in the analytical NTK form. For example, in your citation [1], see equation (37) in the appendix. If $\Theta$ was diagonal, that would require the covariance of $u$ and $v$ in (37) to always be 0 when $u\neq v$, however this is not true in general. For a particular instantiation, see the 2-layer MLP NTK in [3], equations (3-5). You can also use the neural-tangents repository [2] to compute the exact NTK for an arbitrary network, and in general this is not diagonal, let alone $sI$. Perhaps you can clarify what you meant in this particular instance, however I believe the NTK example is misleading and your paper is quite strong without it, and it should be removed in the final draft.
> >
> > [2] Novak et al. Neural Tangents: Fast and Easy Infinite Neural Networks in Python. 2020
> > [3] Bietti and Mairal. On the Inductive Bias of Neural Tangent Kernels, 2019

---

> > > ### Author Response · Authors · 2021-09-02
> > > **Reply to the response**
> > >
> > > Thank you very much for your kind response to our rebuttal, in particular being very positive towards our paper! We will follow your suggestion and remove this example from the manuscript. We will focus more on the core idea and present it better to be a stronger paper. We appreciate your time and patience in making this work impeccable. Thanks! :)

---

### Decision · Program_Chairs · 2021-09-27

**Decision:**

Accept (Poster)

**Comment:**

Many current architectures use "implicit" layers where a solver (e.g. root finder or ODE solver) gives the output. Exactly differentiating these models requires implicit differentiation in the backprop step, which is expensive. This paper introduces a novel "phantom" gradient which is easy to compute, and shows that the inner-product of this gradient with the true gradient is >0, meaning it is a descent direction. An empirical verification shows that optimization using this gradient succeeds.

Reviewers generally felt the paper was novel, well-organized, convincing, clear, and had experiments that show the method truly works. There were a few weaknesses that were identified. First, there were some technical issues in the theoretical results. From the author response, these appear to be relatively minor and specific corrections have been identified. Second, there were some concerns about the rigor of the experiments (e.g. the lack of error bars in the small-scale experiment). Third, there were some concerns about notation and related works. For these latter two concerns the authors have responded at length in the feedback, and this has improved reviewer confidence about these issues. We trust the authors will integrate this content into the main paper, which will strength it considerably.